# Benchmarking and Validation of a Bioinformatics Workflow for Meat Species Identification Using 16S rDNA Metabarcoding

**DOI:** 10.3390/foods12050968

**Published:** 2023-02-24

**Authors:** Grégoire Denay, Laura Preckel, Henning Petersen, Klaus Pietsch, Anne Wöhlke, Claudia Brünen-Nieweler

**Affiliations:** 1Chemical and Veterinary Analytical Institute Rhein-Ruhr-Wupper (CVUA-RRW), Deutscher Ring 100, 47798 Krefeld, Germany; 2Chemical and Veterinary Analytical Institute Muensterland-Emscher-Lippe (CVUA-MEL), Joseph-Koenig-Strasse 40, 48147 Muenster, Germany; 3Chemical and Veterinary Analytical Institute Ostwestfalen-Lippe (CVUA-OWL), Westerfeldstrasse 1, 32758 Detmold, Germany; 4State Institute for Chemical and Veterinary Analysis Freiburg (CVUA-FR), Bissierstrasse 5, 79114 Freiburg, Germany; 5Food and Veterinary Institute, Lower Saxony State Office for Consumer Protection and Food Safety (LAVES), Dresdenstrasse 2, 38124 Braunschweig, Germany

**Keywords:** DNA metabarcoding, amplicon sequencing, food authenticity, food adulteration, next generation sequencing, bioinformatics, validation, benchmarking

## Abstract

DNA-metabarcoding is becoming more widely used for routine authentication of meat-based food and feed products. Several methods validating species identification methods through amplicon sequencing have already been published. These use a variety of barcodes and analysis workflows, however, no methodical comparison of available algorithms and parameter optimization are published hitherto for meat-based products’ authenticity. Additionally, many published methods use very small subsets of the available reference sequences, thereby limiting the potential of the analysis and leading to over-optimistic performance estimates. We here predict and compare the ability of published barcodes to distinguish taxa in the BLAST NT database. We then use a dataset of 79 reference samples, spanning 32 taxa, to benchmark and optimize a metabarcoding analysis workflow for 16S rDNA Illumina sequencing. Furthermore, we provide recommendations as to the parameter choices, sequencing depth, and thresholds that should be used to analyze meat metabarcoding sequencing experiments. The analysis workflow is publicly available, and includes ready-to-use tools for validation and benchmarking.

## 1. Introduction

Commercial food and feed are subjected to international regulations, ensuring that they are safe and conform to the packaging declarations. Meat products are especially prone to adulteration. This can be the replacement of expensive ingredients with cheaper meat products, misinformation by the addition of undeclared components, or the absence of declared components [1,2]. Classical DNA-based methods such as Polymerase Chain Reaction (PCR) amplification, restriction fragment length polymorphism, or DNA-chips, as well as protein-based methods such as ELISA are limited by their target-based approach. As such, their results are limited to a binary answer regarding a single component and they may not be able to identify all ingredients present in a sample. Sanger sequencing, on the other hand, is a widely used untargeted method for the identification of food ingredients. Unlike targeted methods, untargeted methods do not require prior knowledge of specific targets and can analyze a broader range of ingredients. However, it should be noted that the Sanger sequencing application is limited to pure samples [1]. MALDI-based methods are being developed to overcome these challenges, but the collection of reference spectra is still a work-intensive process [3]. Next-generation sequencing (NGS) methods for food authenticity have been developed in the last decade, taking advantage of the untargeted possibilities of the technology and of existing extensive databases of nucleotide sequences [4,5]. 

Some NGS methods focus on a metagenomics approach, i.e., sequencing of all DNA-sequences in a sample [6,7]. Other methods use a metabarcoding approach, in which a small conserved DNA fragment is amplified and sequenced, while sequence differences allow for specific taxa identification [8,9,10,11,12,13,14]. This method allows untargeted species identification and increased parallelization of sample processing and analysis, while taking advantage of the massive amount of reference sequences available in dedicated databases. The choice of barcode is however non-trivial and can have a strong effect on the method’s performance: barcodes should be short enough to still be detectable in highly degraded samples, while still allowing to distinguish between closely related organisms [15]. Metabarcoding presents several advantages over metagenomics: decreased costs, larger reference collections, and less complex analysis, however at the cost of lower taxonomic resolution, being limited to a subset of the taxonomy, and being prone to PCR artifacts [16]. For these reasons, metabarcoding methods are currently widely used for food authenticity determination in a wide-range of matrices [8,9,10,11,12,14,17,18,19].

Various methods for sample preparation and sequencing of meat products were validated and published, focusing on the two main short-read sequencing technologies [8,10,11,13,14]. While the IonTorrent platform offers proprietary data analysis solutions, a number of alternative bioinformatics workflows are published for Illumina sequencing data [8,9,11,14]. However, currently published workflows present various drawbacks: (1) none of them is freely available beyond the publication; (2) parameter choices in these workflows appear to be arbitrary, with no comparison of different parameters and/or tools, and can widely differ between workflows; (3) most validations were performed using a very limited subset of reference databases, yielding over-optimistic validation performances.

Our goal here is three-fold. Firstly, we aim to assess the possibilities of using large databases such as the NCBI NT database [4] for metabarcoding analysis and compare predicted performances of different metabarcoding methods. Secondly, our goal is to benchmark a selection of algorithms and parameter sets and validate an optimized analysis workflow. To this end, we used a dataset of 79 real samples, spanning 32 individual taxa. We methodically optimized the bioinformatics analysis on this dataset and present a set of parameters for optimal analysis performance. Lastly, we calculate the accuracy of the analysis and formulate recommendations regarding the limit of detection and sequencing depth. Both the dataset and software programs used in this study are made freely available to help future improvement and practical applications in food authenticity analysis laboratories.

## 2. Materials and Methods

### 2.1. Reference Material

The 79 reference samples used in the study were acquired from commercial providers DLA Proficiency Tests GmbH (www.dla-lvu.de; accessed on 1 December 2022), Laborvergleichsuntersuchungen Gbr, (www.lvus.de; accessed on 1 December 2022), and LGC Standards (www.lgcstandards.com; accessed on 1 December 2022) were part of interlaboratory ring trials [10], or prepared from certified reference materials or materials whose identity was determined by a certified veterinarian and Sanger sequencing (see Appendix A). Some of these samples were used in a previous study [9].

Samples were prepared and sequenced as previously described [9]. Raw sequencing data are deposited to the European Nucleotide Archive with Project accession PRJEB57117.

Down-sampling was performed using the SeqTK ‘sample’ tool [20]. The sample size and replicate number were concatenated and used as a seed for the random read selection process. The same seed was used on forward and reverse reads.

The BLAST NT database was downloaded on 12 November 2021 and the tax dump files on 19 November 2021 [4].

### 2.2. In Silico Barcode Analysis

Recovery and analysis of the barcode sequences from the BLAST NT Database [4] was performed using the BaRCoD v1.1.1 pipeline [21]. Briefly, the nucleotide database was filtered to include only Amniote sequences. Primer sequences (Appendix A) were searched using the BLAST+ command line tools, with parameters reproducing the implementation of the Primer-BLAST tool. For this, we used a coverage value of 80% and an identity value of 65% [22,23]. Sequences flanked by facing primer sequences were considered barcodes and extracted, and a new BLAST database was created using these sequences. Barcode sequences were then dereplicated in a taxon-wise fashion, primer sequences were removed using cutadapt [24], and global pairwise alignment was performed with VSearch [25]. The pairwise alignments were used to calculate hamming distances. To determine a consensus level for each sequence, we considered all sequences within a given identity level (1-hamming distance/sequence length), the lowest node shared by a majority of taxa was determined as a consensus taxon determined using TaxidTools [26], and the NCBI taxonomy classification [4]. Conspecific probability was calculated as previously described [27].

### 2.3. Metabarcoding Analysis

Sequencing data analysis was performed with FooDMe v1.6.3 [28]. Parameters indicated thereafter were used if not specified otherwise in the text or figures. 

#### 2.3.1. Reads Preprocessing

Primer sequences (Appendix A) and their reverse complements were trimmed from the 5′ and 3′ ends of the reads, respectively, using cutadapt [24] with an error rate of 0.1. Trimmed reads were filtered with fastp [29] to discard reads shorter than 50 bp and trim trails using a window of 4 bp with a minimal quality of 25.

#### 2.3.2. De Novo Identity Clustering

Identity clustering was performed with VSearch [25]. Reads were merged with the ‘--fastq_mergepairs’ function, and a quality filter was applied to keep pseudo-reads between 70 and 100 bp and a maximum of 2 expected errors. Pseudo-reads were dereplicated before being clustered with the ‘--cluster_size’ function, using identity levels between 0.97 and 1.0 (dereplication), and OTUs were sorted by size, discarding clusters with less than 2 reads. If required, chimeras were detected and removed using the ‘--uchime_denovo’ function.

#### 2.3.3. Denoising

Denoising was performed with DADA2 [30]. Read pairs were filtered to remove those with more than 2 expected errors using the ‘filterAndTrim’ function. Forward and reverse error rates were determined with the ‘learnError’ function, and reads were corrected using the error model in the ‘dada’ function. Finally, corrected reads were merged with the ‘mergePairs’ function while allowing for 1 mismatch. If necessary, chimeras were detected and filtered using the ‘removeBimeraDenovo’ function using the ‘per-sample’ method.

#### 2.3.4. Taxonomic Assignment

A mask was created for the BLAST NT database [4] by filtering sequence ID corresponding to Vertebrate taxa. Sequences corresponding to extinct taxa were then filtered from this list. OTUs or ASVs were then searched against the masked database using the BLAST+ program [22] using ‘megablast’ searches with filters for e-value (1.0 × 10^−10^), identity (97), and coverage (100). Results were then filtered by applying a bitscore filter [31] of 4, meaning that for each OTU/ASV, matches with a bitscore difference to the best match for this cluster above 4 were discarded. The consensus taxon for each cluster was determined with TaxidTools [26] by applying a majority vote on the matching taxa, with a minimum threshold of 0.51. The consensus corresponds to the lowest node common to at least X fractions of the taxa, X being the consensus level [32].

### 2.4. Performance Analysis

Run performances were determined using the ‘benchmark’ module of FooDMe [28]. The observed compositions of each sample were compared to their expected values (Appendix A). For this, a concentration threshold of 0.1% was applied and correspondences between expected and predicted values were considered at the genus level. Precision scores, recall scores, average precision scores, and F-scores were calculated using the appropriate functions of the ‘scikit-learn’ or ‘yardstick’ libraries [33,34]. Euclidean distance was determined with NumPy’s ‘linalg.norm’ function [35]. Relative error was determined as $E\;=\;\left|predicted\;-\;expected\right|\;÷\;expected$.

### 2.5. Figure Preparation and Statistical Analysis

Figures were prepared in R using the ‘tidyverse’, ‘ggpubr’, ‘rstatix’, ‘yardstick’, and ‘cowplot’ libraries [34]. Variations within groups were analyzed using the Kruskal-Wallis test. Variations between groups were analyzed using ANOVA on quantile-normalized values (Average precision) or original values (Distance). Yield, average precision, and distance distribution were compared using the Wilcoxon test and *p*-values were corrected for multiple comparisons using FDR correction. Different levels of *p*-values threshold are indicated as follows: n.s. (*p* ≤ 1); * (*p* < 0.05); ** (*p* < 0.01); *** (*p* < 0.001); **** (*p* < 0.0001). 

## 3. Results

### 3.1. Barcode Specificity

Successful identification of the taxon associated with each barcode depends on both the availability of the sequences in the reference database and their differentiability from sequences associated with other taxa. Several distinct methods have been published for birds and mammals barcoding (Table 1 and Appendix A) [8,11,14,36,37,38], targeting different conserved genes: the 16S ribosomal small subunit (16S), Cytochrome B (cytB), or Cytochrome oxidase 1 (COI/COX1). The 16S rDNA-based metabarcoding method published by Dobrovolny et al. in 2019 [11] is currently being adopted as an official method by the German consumer protection authorities and its performances were carefully measured in a recent series of studies [9,10,11,39]. We, therefore, chose to focus our benchmarking and optimization efforts on this method. Nevertheless, we wanted to compare the predicted performances of this method to other published barcoding methods, as potential shortcomings could be overcome by an alternative barcode.

For this purpose, we first extracted all Amniota (the clade grouping birds and mammals taxa, as well as reptiles) barcode sequences for seven different primer sets using a local implementation of the Primer-BLAST algorithm [21,23]. For each different barcode, we determined all other barcodes within 97% identity distance. We then determined the taxonomic assignment consensus that would result in either a 97% or 100% identity clustering for this sequence, based on a strict majority consensus of all barcodes clustering together at the identity level (Table 1). 

Of over 35,000 Amniota taxa represented in the BLAST NT database, the methods based on cytochrome B amplification were the ones that yielded barcodes for the most taxa (over 10,000), with the VDLUFA method yielding barcodes for over 18,000. The COX1/COI methods, on the other hand, were the most restricted, yielding barcodes for under 5500 taxa. All methods yielded high assignment quality for both 97% identity and 100% identity, with more than 95% and 97% of barcodes being assigned at the genus level or below, respectively. The 16S-based method published by Dobrovolny et al. (2019) [11] performed significantly worse at the 97% identity level, with 10% fewer barcodes assigned at the genus level or better, and only slightly worse at the 100% identity level. This might be because the amplicon sequence for this method is especially short (~75 bp excluding amplification primers).

Because most taxa in the BLAST NT database are not relevant for food authenticity, we looked closer at a list of food- and feed-stuff-relevant or -adjacent species curated by the German Consumer Protection and Food Safety Office [40,41]. We examined whether each barcoding method could retrieve at least one barcode for each mammal and bird species in this list (Table 2).

Surprisingly, the 16S and the VDLUFA-cytochrome B methods performed better on the selected species than all other methods. This is likely due to the higher conservation of the primer-binding regions for these methods across the birds and mammals classes. Notably, no methods could find barcodes corresponding to *Anser rossii* (Ross’ goose) and *Marmota marmota* (alpine marmot), which might reflect the absence or bad quality of mitochondrial sequences for these species in the database. The 16S methods were unable to retrieve barcodes for *Dama dama* (fallow deer) and *Equus quagga* (plains zebra), whereas the VDLUFA method was able to. The inability of the Dobrovolny method to amplify fallow deer was reported before [9] and the method has since then been improved to overcome this problem [42]. 

Aside from that, the 16-based Dobrovolny method [11] features several advantages. Firstly, the very short amplicons make this method the most suitable for highly processed matrices, where DNA might be heavily degraded [15,43]. Secondly, its ability to amplify most species of interest (Table 2) and a large spectrum of Amniota taxa (Table 1) was shown in silico. Thirdly, there is a growing body of literature on the suitability of this method for metabarcoding experiments, including interlaboratory validation [9,10,11]. This method also presents some drawbacks, namely comparatively bad predicted performances at 97% identity clustering, and the species rank for 100% identity clustering (Table 1). However, for meat speciation in food and feed, identification at the genus level is usually sufficient for the detection of fraud.

### 3.2. Workflow Benchmarking and Optimization

The metabarcoding data analysis workflow can be separated into three main phases: Reads preprocessing, where primer sequences are removed, and bad quality bases are trimmed;Clustering, where reads satisfying a given identity level are grouped together;Taxonomic assignment, where clusters of reads are assigned to taxonomic nodes.

Each of these steps can be performed using a variety of different algorithms, each with several parameters, whose values can have a strong impact on the quality of the analysis. Several studies of meat-products metabarcoding have been published in the past years, each using different analysis workflows and reference databases (Appendix A). In order to objectively compare different algorithms and parameters, we collected 16S metabarcoding experiments for 79 different samples, totalizing 32 different species (Appendix A). The dataset is enriched for taxa at around 1%, which is the threshold commonly used in diagnostic laboratories as a lower limit for legal action in case of undeclared species. For each parameter set, we analyzed all samples assigning reads using the full BLAST NT database. The workflow’s performances were determined both qualitatively and quantitatively. Qualitatively, we compared the observed and expected composition of the samples at the genus level, and we calculated the average precision, which is the geometric mean of the precision and recall. Quantitatively, we determined the yield of the analysis as the number of reads retained through to taxonomic assignment and calculated the Euclidean (or L2) distance between the vector of predicted values and the vector of expected values, reflecting how far predictions are from the expected compositions of the samples.

#### 3.2.1. Benchmarking Clustering Parameters

The main bottleneck of the analysis is to obtain read clusters that accurately describe the real composition of the sample. Several clustering methods have been described, amongst which de novo identity clustering and denoising are the main representatives [30,44,45,46]. We here compared the effects of clustering reads with a 95%, 97%, and 100% (dereplication) identity threshold and using denoising (Figure 1).

While the clustering algorithm choice made no significant difference in the qualitative and quantitative accuracy of the results (Figure 1A,B), the denoising method had a significantly higher yield than de novo clustering with either identity level (Figure 1C). The median yield for denoising was 99%, with most samples above 98%, whereas it fell to a median yield of 97% for dereplication, with samples as low as 93% yield. Additionally, the clusters produced by the denoising algorithm were much closer to the real sample composition, as shown by the splitting level, calculated by taking the log10 of the ratio of cluster number to expected components in each sample (Figure 1D). The splitting level median was close to one for denoising, indicating that each “real” sequence was split between 10 clusters. This level was increasingly high with increasing identity level for the de novo identity clustering method. It reached a median value of 2.5 for the dereplication, indicating that each “real” sequence was split between over 300 clusters. This had a significant impact on the run time of the workflow (Figure 1E). The dereplication method ran over 20 min per sample per core, whereas other methods ran for under 10 min per sample per core. This is consistent with previous work showing a more accurate clustering using denoising [27,45].

When using denoising, we noticed that it was important to allow for one mismatch for reads merging (Appendix A). Using a strict identity for merging resulted in a considerable yield loss of up to 8%, while allowing one mismatch did not affect the quantitative and qualitative accuracy of the results while maximizing yield. This is due to the short size of the amplicon, which combined with a long read length means that the entire barcode (~75 bp) is used as an overlapping sequence for merging.

We also checked whether detecting and filtering chimeric reads after denoising influenced the results. Although filtering chimera slightly affected yield, in the order of a few percent of the reads, neither the quantitative nor the qualitative performances of the workflow were affected (Appendix A). This likely indicates that very few chimeric sequences are formed during both PCR and sequencing steps using the previously published 16S method [9,11].

In conclusion, we show here that all four tested clustering algorithms yield similar results. However, using denoising while allowing one mismatch in the overlapping sequences during read merging maximizes yield and gives clusters closer to the expected sample composition.

#### 3.2.2. Optimization of Taxonomic Assignment

Consensus sequences for each cluster need to be assigned to a taxonomic node. This was done using the ‘megablast’ tool by looking for highly similar sequences in the BLAST NT database. As only part of the database is relevant for the identification of birds and mammals, the database was pre-filtered to exclude taxa not belonging to the Vertebrate clade. A BLAST search typically yields many results, most of which are far off the target. In order to narrow the search, several filters are available [31]. We applied a first hard filter consisting of an expect-value (E-value; describes the number of hits one might expect to see by chance in the database) and an identity level (the fraction of identical nucleotides between hit and query) thresholds. Results were then post-filtered using bitscore difference to keep only results within a certain distance to the best result for this query. Finally, each cluster was assigned to a taxonomic node based on a minimal consensus level, between 51% (majority consensus) and 100% (last common ancestor). Using this process, it is possible to assign a taxonomic node to all clusters, even with divergent results, although the consensus may be at the genus or higher rank [32]. 

The E-value threshold did not influence the assignment accuracy (Appendix A). This is due to the downstream decision of considering only the top results from the BLAST search. It is, however, important to note that using an E-Value threshold lower than 1.0 × 10^−20^ returned no hits from the BLAST search.

Because the multiple filtering process can have complex synergistic effects, we used a matrix design to test a range of values for each filter: BLAST identity level was varied between 95% and 100%, bitscore difference between 0 and 8 bits, and minimal consensus between 51% and 100%. We then checked if any filter, or combination of filters, had a significant effect on the result’s accuracy using analysis of variance (ANOVA).

The ANOVA of average precision values showed a significant effect on each individual filter (Figure 2A). As well as combined effects of the minimal consensus and bitscore filters. However, when omitting the parameter value of 8 for the bitscore filter, which gave significantly worse results than any other (Figure 2C), the interaction effect was not observed anymore. We, therefore, analyzed each effect individually. Average precision improved gradually with improving BLAST identity values, yielding the best results for 100% identity values (Figure 2B). In this context, the bitscore filter was redundant with the identity filter and yielded similar results for any value below 8 (Figure 2C). Consensus gave the best results with values of 80% or below, the value of 100% (last common ancestor) gave significantly worse results (Figure 2D). 

For the Euclidean distance, ANOVA only detected a significant effect of the consensus filter (Figure 2E). Here again, all values below 100% yielded similar results (Figure 2F). The loss of quantitative accuracy at 100% is most likely due to the decrease in qualitative performance at this level, leading to a misassignment of sequences, ultimately resulting in a different predicted composition.

Based on these results, the taxonomic assignment appears very robust to different filtering values within the measured ranges. Best results are observed with a BLAST identity value of 100%, the bitscore difference filter should allow for a maximum of 4 bits difference and consensus should be determined using a majority vote (51%). More stringent parameters (lower bitscore difference and higher minimum consensus level) could be used if necessary to distinguish highly similar sequences, without predicted adverse effects.

### 3.3. Detection Limit

A common strategy to filter noise from real signals is to set a minimal proportion threshold under which the signal is considered negative. To find the optimal threshold, we calculated precision, and recall in 0.01% of total composition increments (Figure 3A). Recall rapidly decayed after 1%, consistent with the fact that many components were present at a 1% proportion in the dataset. Precision rapidly increased before reaching a plateau at around 0.1%. To find the optimal threshold, we calculated the F2-score, which is the geometric mean of the recall and precision, whereby the recall is considered twice as important as the precision. The F2-score maximum was reached at a threshold value of 0.093%, which can be rounded to 0.1%. This threshold value agrees with the previously published values for small curated databases [10,11].

### 3.4. Performance Evaluation

Based on the choice of parameters and threshold exposed previously, we calculated various performance metrics for the workflow at both species level, which is the highest resolution that can reasonably be obtained, and genus level, which, although not as resolutive as species, generally yields sufficient information for authenticity determination. At the species and genus levels, respectively, we observed a precision of 90% and 98%, meaning that 10% and 2% of the determined taxa were not expected in the samples. We also observed a recall of 93% and 96%, respectively, meaning that 7% and 4% of the taxa present in the sample were not found (Table 3). These results include 18 samples containing fallow deer (*Dama dama*) to various levels (Appendix A). As we showed earlier, and as was previously published, fallow deer is a known miss for the 16S Dobrovolny method used here [9]. When correcting for fallow deer, the precision and recall increased to 90% and 96% at the species level, and 98% and 99% at the genus level. This is slightly lower than the previously reported 100% precision and recall for the method. However, these reports were based on the use of a custom database containing 51 entries at most [9,10,11]. The database used here contains entries for 16S sequences of over 7700 taxa (Table 1). It should be noted that the performance values reported here are slightly under-estimated. This is due to the fact that some proficiency test samples contain trace amounts of species not added intentionally, e.g., LVU_2018_B, DLA45/2019-2, and DLAptAUS2/2020-3.1, where the majority of participants detected red deer, goat, and horse, respectively, in addition to the expected species [47]. Similarly, several prediction errors are likely linked to incorrect sample compositions: both replicates of the LGC 7244 samples are false negative for chicken, due to chicken being detected under the 0.1% threshold, which was also reported in a previous study with another bioinformatic method [9]. The same study reported a goat positive result for the unintentional traces of goat in sample DLA45/2019-2, which we also observed with our method.

We also measured the quantitative performance of the workflow by comparing expected vs. predicted proportions of components in the samples. For this, we calculated the absolute value of the difference between expected and predicted proportions and normalized it by the expected proportion (Figure 3B). The relative quantification error peaked at about 60% for components present at 1% in the samples and decreased to a few percent for components making up more than half of the sample. However, a large variance was observed, and the relative error varied to up to five times the expected amount for some low-concentration components. These values are within the variance reported previously, which was shown to be comparable to quantitative real-time PCR assays [9].

In conclusion, the workflow presented here is a very robust screening method for detecting components at 1% or higher. Results should however be interpreted with care and confirmed with a parallel assay such as quantitative or digital PCR, in particular, if quantification is needed.

### 3.5. Effects of Sequencing Depth on Prediction Recall and Variance

To determine the effects of sequencing depth on the precision and robustness of the results, we selected a subset of 35 samples with at least 350,000 read-pairs each, and whose composition structure was similar to that of the full dataset (Appendix A). We then randomly selected subsets of the samples to produce down-samples with 1000 to 200,000 reads each, in 8 independent replicates. Recall and Euclidean distances were then determined for each down-sample as previously described.

Apart from the sample with 1000 reads, all sampling depths led to comparable results in terms of recall and variance thereof (Figure 4A,B). With only 1000, we could observe a drop in the recall in many samples, which was associated with an increase in the variance of the recall within sample replicates. Only marginal improvements could be observed above 5000 reads, and until 80,000 reads. Above 80,000 reads, no increase in recall could be measured. The Euclidean distance, however, did not vary across the measured range of sampling depths (Figure 4C).

The performance plateau reached at 5000 reads is consistent with a previous study on environmental samples metabarcoding [48], while theoretical calculations for metagenomics experiments proposed 15,000 reads for taxa identification at a 1% threshold [6]. These values are however results of simulations and should be verified experimentally. Most importantly, these do not account for random noise, which might become more visible when sequencing at low depths.

## 4. Discussion

Our aims here were threefold: (1) Compare the suitability of different published barcode sequences to distinguish mammal and bird species in the full BLAST NT database (2) Optimize the data analysis workflow for Illumina 16S rDNA sequencing using a large dataset of reference samples representative of typical food samples; (3) Validate the workflow and provide minimal input data criteria for accurate and reproducible analysis. 

The choice of the barcode for metabarcoding analysis is not trivial and can heavily influence the results of the experiments. One must consider how common this sequence is in the taxa of interests, whether it is specific enough to distinguish different species, and whether it can reliably be amplified from the samples. We here compared a selection of 7 different published barcodes used for either Sanger-based species identification [36,37,38] or in metabarcoding experiments [8,11,14]. These sequences target either the 16S rDNA, COI/COX1, or cytB genes. For each of these, we determined how many taxa could be retrieved from the BLAST NT database. We measured the taxonomic specificity of each sequence retrieved this way. Finally, we specifically checked whether a set of usual food components or contaminants [40] could be identified using these sequences. We observed that cytB-based methods and especially the VDLUFA method [38], had the best-predicted performances across the Vertebrate clade. The 16S method from Dobrovolny et al. suffered from a lower predicted accuracy but was expected to be able to amplify sequences from most taxa of interest. In addition, it has the advantage of using a very short sequence of ~75 bp, allowing the analysis of highly processed products in which nucleic acids may be degraded [15]. This method is also currently being validated on a large scale and was previously shown to work for a range of food and feed products [9,10]. For these reasons, we chose to focus on this method for the rest of the study. It should however be noted that we predicted the cytB-based VDLUFA method to perform also very well, amplifying and distinguishing a large range of species. It could be a good complement to the Dobrovolny method, provided that the longer sequences of ~220 bp can be amplified from the samples to analyze. 

We then set on optimizing the data analysis workflow by comparing different algorithms and parameter combinations on a set of reference samples. These span 32 taxa of interest and are enriched for concentrations at around 1%, which is the limit usually used for legal labeling obligations. The data analysis workflow can be separated into three main steps. Firstly, the reads are checked for quality, amplification primers and sequencing adapters are removed, and bad quality trailing bases are trimmed. Then the reads are grouped into clusters of similar sequences. Finally, each cluster is assigned to a taxonomic node [49]. With the selected method, the barcode is significantly shorter (~75 bp) than the sequencing reads (150 bp). In this case, the sequencing read goes through the amplicon. After primer trimming, the reads are halved, and trailing bad quality bases are already removed. Optimization of the read trimming was, therefore, not necessary here. This should, however, be considered when using a longer barcode, or when sequencing with shorter read lengths available on Illumina platforms.

The clustering step is important as it determines which sequences will be used to interrogate the reference database, and ultimately determine sample composition. Classically, reads are clustered in Operational Taxonomic Units (OTUs) by identity threshold, in general with an identity level of 97%, although it was discussed in recent years that much higher thresholds should be used [27,46]. The accuracy of OTUs is now largely contested, and new statistical methods have been published that aim at determining Amplicon Sequence Variants (ASVs) using denoising procedures [30]. ASVs are expected to better represent the sample composition, whereas OTUs generally overestimate diversity [45]. While OTUs at 97% identity group highly similar sequences and 100% identity would simply dereplicate sequencing reads without filtering noise, ASVs address both issues by determining the correct sequences based on read quality scores. However, several publications nuanced this claim, putting ASVs on par with OTUs [50,51]. Here, we compared the effects of identity clustering at 95%, 97%, 100% (dereplication), and denoising and found no significant differences in terms of prediction accuracy. ASVs did result in a slightly higher yield, and the number of clusters determined was much closer to the true composition of the sample, resulting in a faster processing time than dereplication. For these reasons, we chose to keep working with ASVs, although OTUs led to highly similar results. We expect this result to be generalizable to different barcodes.

Most published meat-metabarcoding methods used reference databases containing only a selected number of entries (between 2 and 500). This has the advantage of ensuring that the database only contains high confidence, high quality sequences and drastically simplifies taxonomic assignment. However, this has the drawback of hiding a large chunk of potential adulteration with species not present in the reference database. Taxonomic assignment using the full BLAST NT database as a reference revealed challenges due to mislabelling of sequences, the presence of low quality sequences, and large heterogeneity in taxa representations. We used a set of filters that enabled us to overcome these hurdles. We restricted the BLAST search by using a mask selecting taxa placed under the Vertebrate node, thus ensuring that no other sequences would contaminate the results. Furthermore, we then applied a coverage threshold of 100%, ensuring that only sequences aligning to the totality of the barcode would be recovered. The minimal identity level required was varied between 95% and 100%, and we determined that it should optimally be 98% or higher. We then used a filter based on the bitscore value of each result, representing the quality of the alignment between the barcode and the reference. Although the E-value is commonly used for filtering BLAST results, this value varies with the size of the database, rendering an E-value filter obsolete as the database grows in size, whereas the bitscore only depends on the alignment and therefore only varies with the barcode used [52]. Therefore, we determined the bitscore of the best alignment for each barcode sequence and kept only these results that were within a certain distance of the best results. We found the optimal value for the bitscore difference to be between 0 and 4. These filtering procedures typically resulted in a few different possible taxa. In order to determine the most likely taxon for each sequence, we used a consensus threshold [32] and found the optimal value of the minimal consensus to be between 51% and 80%. This allowed the assignment of a unique taxon to each barcode sequence in the sample, albeit at a rank depending on the confidence of the BLAST result. Most results were assigned at the species or genus level, allowing for a meaningful interpretation of the results. We expect the parameter choices for the taxonomic assignment step to be highly dependent on both the database and the barcode used. They should therefore be revalidated for each barcode and when the database is updated.

Errors stemming from the PCR-amplification and sequencing processes almost always result in a low amount of wrongly assigned reads that need to be filtered. Fixing a threshold at which results become significant becomes increasingly difficult as the detection limit decreases. We here determined the best threshold to be at around 0.1% to optimally balance precision and recall of the analysis, while ensuring a detection limit at 1% of the true sample composition. The optimized workflow resulted in 1% of both false-positive and false-negative results, while the quantification accuracy was comparable to that of other DNA-based methods [9]. Moreover, the results were identical to those of a previously published analysis of part of this dataset using a different bioinformatics workflow and a database consisting of only 51 entries [9]. These validation values indicate that the workflow allows for robust screening methods. The cost-effectiveness of NGS-based methods is an important consideration for laboratories. It was previously calculated that metabarcoding can be more cost-efficient than PCR when parallelizing enough samples [9]. In order to estimate the number of reads required for a robust analysis, we down-sampled a part of our dataset and calculated the recall and variability of the analysis for a range of 1000 to 200,000 reads. We determined that the results did not significantly improve beyond 5000 reads. This is far below the previous recommendations of 120,000 to 200,000 reads [9,11]. There is therefore a potential for further decreasing the costs of metabarcoding analysis. This result is in line with previous calculations [48], but should be verified experimentally.

The use of the full BLAST NT database allows us to take full advantage of the untargeted aspect of metabarcoding methods, however, it comes with its own set of limitations as the database contains some doubtful sequences and annotations and is highly biased towards commercially interesting and model species. One could consider using the BOLD database, which is more tightly curated than BLAST [5]. This database is, however, largely incomplete for 16S rDNA sequences, and more suitable for cytB or COI/COX1 sequences. Several publications have tried to address this problem directly, either by selecting sequences with trustful metadata and setting some quality filters [27], or by assigning quality scores to each sequence and using them to either filter the database or assign confidence scores to metabarcoding results [53]. These approaches should be explored in future works, with the aim of optimizing food authentication methods.

In order to improve reproducible analysis and ease the dissemination of the metabarcoding methods in laboratories, we packaged the analysis workflow in a free open-source software [28]. The workflow is implemented using Snakemake [54] and runs on UNIX platforms. It allows for scalable and reproducible automated analysis of amplicon sequencing runs on Illumina platforms. It also contains a module comparing the workflow’s results to the theoretical composition of the samples, making the validation process straightforward. Another module allows us to run the same analysis using different sets of parameters and compare the results with each other, significantly simplifying the process of finding the optimal set of parameters for new barcodes or matrices. 

## 5. Conclusions

The results of the present study demonstrate the accuracy and robustness of the 16S metabarcoding method as a tool for meat authenticity testing. This study proposes standardization of the data analysis and shows that it can be used with non-curated nucleotide databases such as the BLAST NT database, thereby expanding the detection range. Furthermore, the workflow presented here is versatile and can be adapted to other food matrices, such as plants, seafood, insects, or spice mixtures, making it a valuable tool for a wide range of products. With the increasing demand for transparency and traceability in the food supply chain, this method has the potential to play a significant role in helping to ensure that consumers can trust the food they are eating.

## Figures and Tables

**Figure 1 foods-12-00968-f001:**
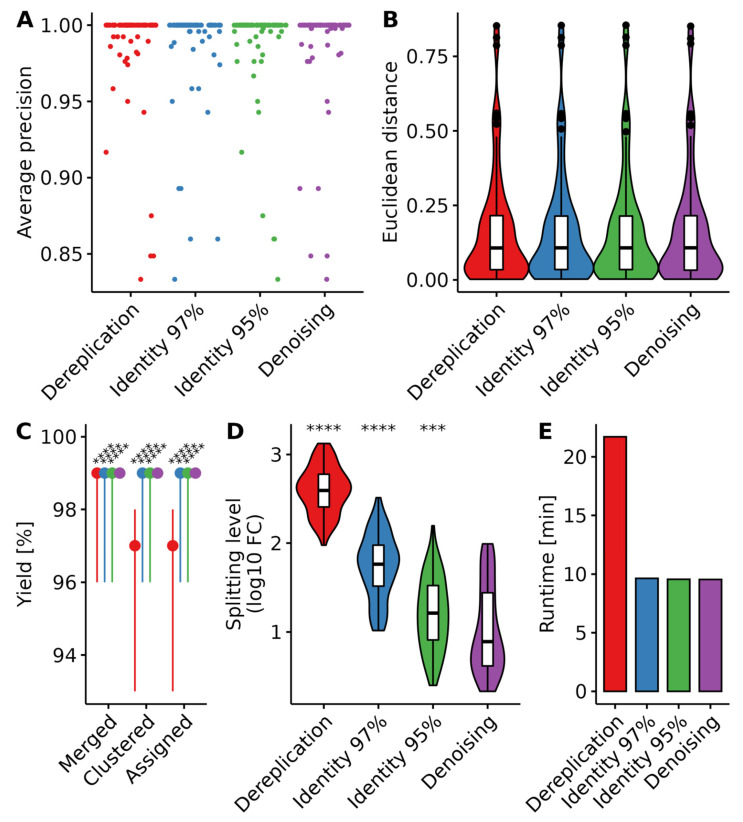
Effects of the clustering method on workflow performances. (**A**) Average precision; (**B**) Euclidean distance between predicted and expected compositions; (**C**) Workflow yields in the number of reads after the read merging step (Merged), after clustering (Clustered), and after taxonomic assignment (Assigned). The dots indicate the median value within each group and the lines represent the range of the distribution. Differences between the groups’ means were tested with the Wilcoxon test for paired samples, using Denoising as the reference group. (**D**) Amplicon sequence splitting level is expressed as the log10-fold change between the expected number of taxa in each sample and the number of predicted sequence clusters. (**E**) Average analysis runtime for the 79 samples dataset, expressed in minutes per core per sample. Violin plot outlines represent the kernel density function of the distribution, the included white boxplots represent the range (lines), quartiles (box edges) and median (middle line of the boxes) of the distribution, and outliers are represented by a black point. Different levels of *p*-values threshold are indicated as follows: *** (*p* < 0.001); **** (*p* < 0.0001).

**Figure 2 foods-12-00968-f002:**
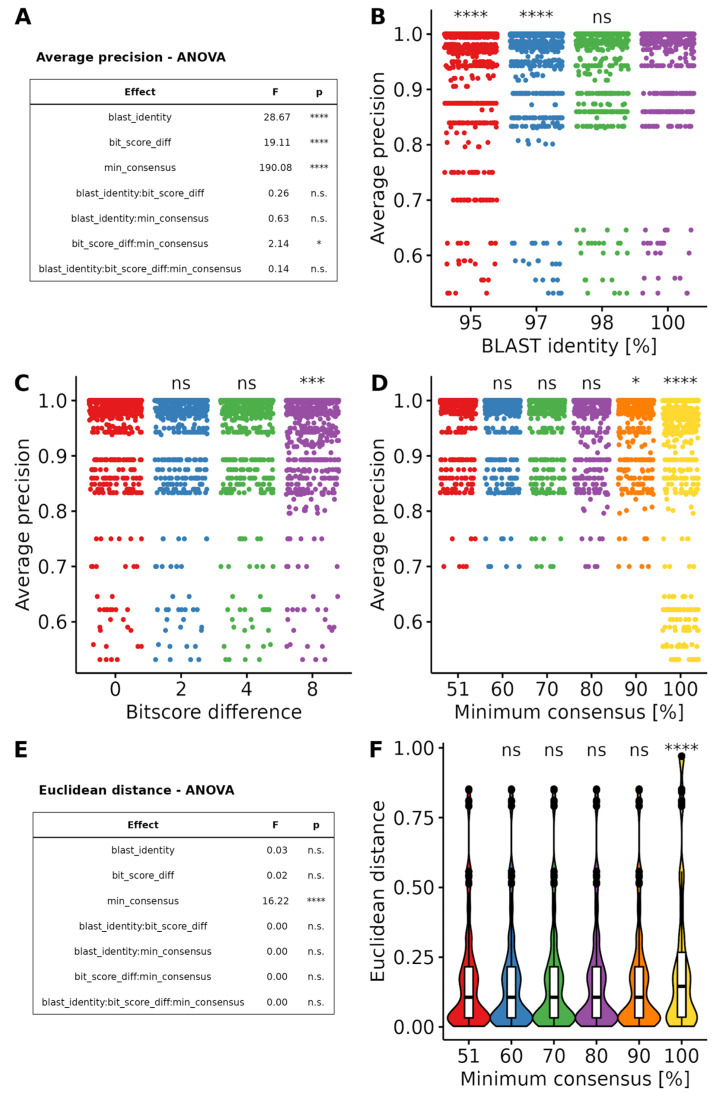
Effects of BLAST filtering parameters on the predictions’ accuracy. (**A**) Summary table of the ANOVA on quantile-quantile normalized average precision values showing F statistic and p-value for single effects and their interactions. (**B**) Average precision for different BLAST identity values. (**C**) Average precision for different bitscore-difference values. (**D**) Average precision for different minimum consensus values. (**E**) Summary table of the ANOVA on Euclidean distance values showing F statistic and p-value for single effects and their interactions. (**F**) Euclidean distance for different minimum consensus values. Violin plot outlines represent the kernel density plot of the distribution, the included white boxplots represent the range (lines), quartiles (box edges) and median (middle line of the boxes) of the distribution, and outliers are represented by a black point. Different levels of *p*-values threshold are indicated as follows: n.s. (*p* ≤ 1); * (*p* < 0.05); *** (*p* < 0.001); **** (*p* < 0.0001).

**Figure 3 foods-12-00968-f003:**
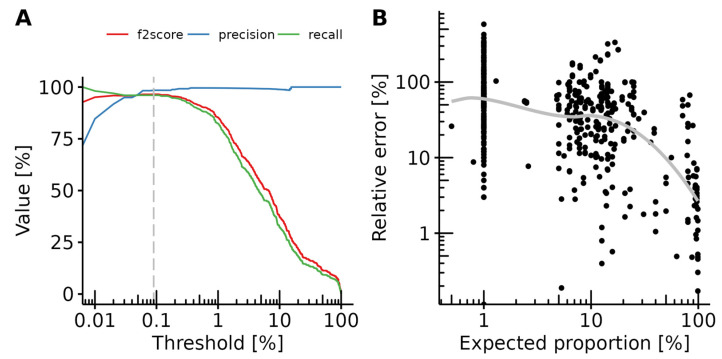
Workflow performances and determination of a minimal composition threshold. (**A**) The precision-recall curve shows the precision (blue), recall (green) and F2-scores (red) over a range of thresholds. The dashed grey line shows the threshold with the maximum F2-score. The *X*-axis is log10 scaled. (**B**) Relative quantification error in the function of the expected proportion in the sample. Each dot represents a single true positive result. The grey line represents the local estimated scatterplot smoothing. Both axes are log10 scaled.

**Figure 4 foods-12-00968-f004:**
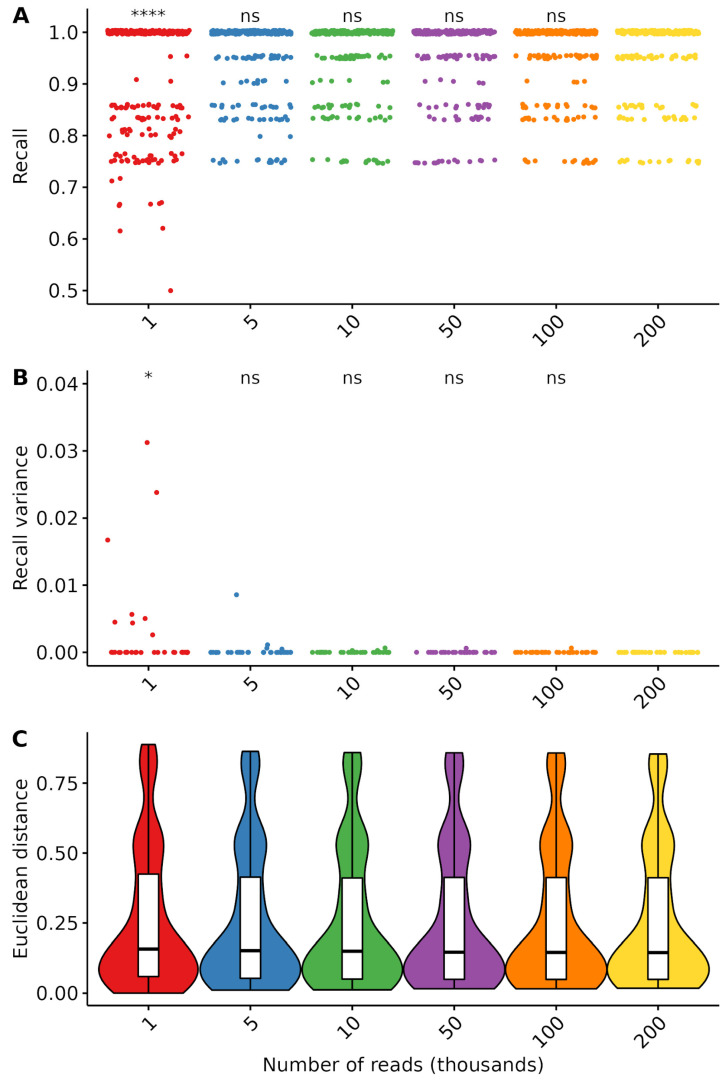
Effects of subsampling of prediction recall and robustness. (**A**) Sample recall across a range of sampling depths from 1000 to 200,000 reads. (**B**) Recall variance across a range of sampling depths. Each dot represents a sample. (**C**) Euclidean distance does not vary in the observed sampling depth range. Violin plots outlines represent the kernel density plot of the distribution, the included white boxplots represent the range (lines), quartiles (box edges) and median (middle line of the boxes) of the distribution, outliers are represented by a black point. Different levels of *p*-values threshold are indicated as follows: n.s. (*p* ≤ 1); * (*p* < 0.05); **** (*p* < 0.0001).

**Table 1 foods-12-00968-t001:** Comparison of barcode number and assignment rank for different targets.

Method	Number of Taxids Retrieved	Median Number of Barcode per Taxid ^a^	Median Length of Barcode [bp] ^b^	97% Identity		100% Identity	
				Sequences Assigned at Max.		Sequences Assigned at Max.	
				Species Level [%]	Genus Level [%]	Species Level [%]	Genus Level [%]
16S_dobrovolny	7701	3	75	66.77	85.36	81.98	93.27
16S_xing	6383	3	204	78.53	95.07	93.29	98.21
cox1_palumbi	2293	5	522	90.16	97.82	97.08	98.63
cytB_meyer	10,909	25	333	91.31	98.47	95.29	98.89
cytB_palumbi	10,078	35	741	94.55	99.17	97.93	99.56
cytB_VDLUFA	18,136	16	220	89.95	98.52	94.38	98.85
miniCOI_palumbo	5482	2	151	82.55	95.46	92.86	97.71

^a^ Unique sequences only. ^b^ Not including primer sequences.

**Table 2 foods-12-00968-t002:** Food- and feed- relevant and -adjacent species amplifiability predictions for different barcoding methods. A ‘+’ indicates that at least one sequence was retrieved for the given organism using the method in the header, and a grayed ‘0’ indicates that no sequence was retrieved for this organism.

Organism	Taxid	Common Name	16S Dobrovolny	16S Xing	COX1 Palumbi	cytB Meyer	cytB Palumbi	cytB VDLUFA	MiniCOI Palumbo
*Addax nasomaculatus*	59515	Addax	+	+	+	+	+	+	+
*Ailuropoda melanoleuca*	9646	Giant panda	+	+	0	+	+	+	+
*Alcelaphus buselaphus*	59517	Hartebeest	+	+	0	+	+	+	+
*Alcelaphus caama*	59519	Red hartebeest	+	+	0	+	+	+	+
*Alces alces*	9852	Eurasian elk	+	+	0	0	0	+	+
*Alectoris chukar*	9078	Chukar partridge	+	+	+	+	+	+	+
*Ammotragus lervia*	9899	Barbary sheep	+	+	+	+	+	+	+
*Anas platyrhynchos*	8839	Duck	+	+	+	+	+	+	+
*Anser anser*	8843	Greylag goose	+	+	+	+	+	+	+
*Anser cygnoides*	8845	Chinese goose	+	+	0	+	+	+	+
*Anser indicus*	8846	Bar-headed goose	+	+	+	+	+	+	+
*Anser rossii*	56281	Ross’ goose	0	0	0	0	0	0	0
*Antidorcas marsupialis*	59523	Springbok	+	+	+	+	+	+	+
*Bison bison*	9901	Bison	+	+	0	+	+	+	+
*Bison bonasus*	9902	Wisent	+	+	+	+	+	+	+
*Bos mutus*	72004	Yak	+	+	0	+	+	+	+
*Bos taurus*	9913	Cattle	+	+	+	+	+	+	+
*Bubalus bubalis*	89462	Water buffalo	+	+	+	+	+	+	+
*Cairina moschata*	8855	Muscovy duck	+	+	0	+	+	+	+
*Canis lupus*	9612	Grey wolf, dog	+	+	+	+	+	+	+
*Capra aegagrus*	9923	Wild goat	+	+	+	+	+	+	+
*Capra hircus*	9925	Domestic goat	+	+	+	+	+	+	+
*Capra ibex*	72542	Ibex	+	+	+	+	+	+	+
*Capreolus capreolus*	9858	Roe deer	+	+	0	+	+	+	+
*Cavia porcellus*	10141	Guinea pig	+	+	0	+	+	+	+
*Cervus elaphus*	9860	Red deer	+	+	+	+	+	+	+
*Cervus nippon*	9863	Sika deer	+	+	0	+	+	+	+
*Columba livia*	8932	Domestic pigeon	+	+	0	+	+	+	+
*Connochaetes gnou*	59528	Black wildebeest	+	+	0	+	+	+	+
*Connochaetes taurinus*	9927	Blue wildebeest	+	+	+	+	+	+	+
*Coturnix coturnix*	9091	Common quail	+	+	0	+	+	+	+
*Coturnix japonica*	93934	Japanese quail	+	+	0	+	+	+	+
*Cygnus olor*	8869	Mute swan	+	+	0	+	+	+	+
*Dama dama*	30532	Fallow deer	0	0	0	0	0	+	0
*Damaliscus pygargus*	9931	Bontebok	+	+	+	+	+	+	+
*Equus asinus*	9793	Donkey	+	+	+	+	+	+	+
*Equus caballus*	9796	Horse	+	+	+	+	+	+	+
*Equus quagga*	89248	Plain zebra	0	0	0	0	0	+	0
*Equus zebra*	9791	Mountain zebra	+	+	+	+	+	+	+
*Felis catus*	9685	Cat	+	+	+	+	+	+	+
*Gallus gallus*	9031	Chicken	+	+	+	+	+	+	+
*Gazella dorcas*	37751	Dorcas gazelle	+	+	0	+	+	+	+
*Gazella subgutturosa*	59529	Black-tailed gazelle	+	+	0	+	+	+	+
*Glis glis*	41261	Fat dormouse	+	+	0	+	+	+	+
*Hippotragus niger*	37189	Sable antelope	+	+	0	+	+	+	+
*Kobus ellipsiprymnus*	9962	Waterbuck	+	+	0	+	+	+	+
*Lama glama*	9844	Llama	+	+	+	+	+	+	+
*Lepus europaeus*	9983	European hare	+	+	0	+	+	+	+
*Macropus fuliginosus*	9316	Western gray kangaroo	+	+	0	0	0	+	0
*Macropus giganteus*	9317	Eastern gray kangaroo	+	+	0	+	+	+	+
*Marmota marmota*	9993	Alpine marmot	0	0	0	0	0	0	0
*Martes martes*	29065	European pine marten	+	+	0	+	+	+	+
*Meleagris gallopavo*	9103	Turkey	+	+	+	+	+	+	+
*Muntiacus reevesi*	9886	Reeves’ muntjac	+	+	0	+	+	+	+
*Mus musculus*	10090	Mouse	+	+	+	+	+	+	+
*Myodes glareolus*	447135	Bank vole	+	+	+	+	+	+	+
*Numida meleagris*	8996	Helmeted guineafowl	+	+	0	+	0	+	+
*Oryctolagus cuniculus*	9986	Rabbit	+	+	+	+	+	+	+
*Oryx dammah*	59534	Scimitar-horned oryx	+	+	+	+	+	+	+
*Oryx gazella*	9958	Gemsbok	+	+	+	+	+	+	+
*Osphranter robustus*	9319	Common wallaroo	+	+	0	+	+	+	+
*Osphranter rufus*	9321	Red kangaroo	+	+	+	+	+	+	+
*Ovibos moschatus*	37176	Musk ox	+	+	0	0	0	+	+
*Ovis aries*	9940	Sheep	+	+	+	+	+	+	+
*Ovis orientalis*	469796	Asiatic mouflon	+	+	+	+	+	+	+
*Phasianus colchicus*	9054	Common pheasant	+	+	0	+	+	+	+
*Rangifer tarandus*	9870	Reindeer	+	+	+	+	+	+	+
*Rattus norvegicus*	10116	Rat	+	+	+	+	+	+	+
*Struthio camelus*	8801	Ostrich	+	+	0	+	+	+	+
*Sus scrofa*	9823	Pig	+	+	+	+	+	+	+
*Syncerus caffer*	9970	African buffalo	+	+	+	+	+	+	+
*Tragelaphus oryx*	9945	Eland	+	+	+	+	+	+	+
*Tragelaphus spekii*	69298	Sitatunga	+	+	0	+	+	+	+
*Vulpes vulpes*	9627	Red fox	+	+	+	+	+	+	+

**Table 3 foods-12-00968-t003:** Qualitative performance summary of the workflow.

Evaluation Rank	Confusion Matrix			Performance Metrics	
	True Positives	False Positives	False Negatives	Recall	Precision
Species ^a^	490 (84%)	56 (10%)	39 (7%)	93%	90%
Species ^b^	490 (86%)	56 (10%)	21 (4%)	96%	90%
Genus ^a^	494 (95%)	6 (1%)	21 (4%)	96%	99%
Genus ^b^	494 (98%)	6 (1%)	3 (1%)	99%	99%

^a^ Including 18 *Dama dama* components. ^b^ Corrected for *Dama dama.*

## Data Availability

The FooDMe and BaRCoD workflows are openly available from github and referenced on Zenodo at: github.com/CVUA-RRW/FooDMe (DOI: 10.5281/zenodo.7078595) and github.com/CVUA-RRW/BaRCoD (DOI: 10.5281/zenodo.6976282). The dataset used for the analysis is openly available from the European Nucleotide Archive (ENA) at ebi.ac.uk/ena/browser/home, under Project accession PRJEB57117. The NCBI NT database is available from ftp.ncbi.nlm.nih.gov/blast/db/. Additional code used for figure preparation and detailed analysis results for all samples and parameter combinations are available upon request.

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
