# Peer review of "Benchmarking and Validation of a Bioinformatics Workflow for Meat Species Identification Using 16S rDNA Metabarcoding"

_foods, 2023, doi:10.3390/foods12050968_

Round 1

Reviewer 1 Report

The research article "Benchmarking and validation of a bioinformatics workflow for meat species identification using 16S rDNA metabarcoding" is a well written article. Making the 'analysis workflow' as open access is appreciable. While I have no major comments conceptually, I have minor comments in the introduction part.

Fourth line of the introduction: "Classical DNA-based methods such as PCR,....'. As authors know that the PCR is a method for amplification, and authors just using the term "PCR' in the sentence can be elaborated.

Similarly, in the sixth line, "Sanger-sequencing is widely used to identify ingredients in an untargeted manner," is confusing. What authors mean by 'untargeted manner' here?.

Authors can provide the number of samples sequenced in this study in the methodology part itself rather than revealing in the results section.

Reviewer 2 Report

The present study presents a workflow and tools with the aim of comparing the suitability of different published barcode sequences to distinguish mammal and birds species, optimizing the data analysis workflow for Illumina 16S rDNA sequencing. The study is interesting.

-How do you explain that 16S and the VDLUFA-cytochrome B methods performed better on the selected species than all other methods?

-Please change the conclusion, maybe discussing the application of this workflow in the food industry. The conclusion written in this way is a summary of the study and not a real conclusion

Reviewer 3 Report

Nowadays there is a strong need to ensure that food products (especially those of animal origin) is safe and conform to the packaging declarations. The problem is important by means of both nutrition and health properties of food as well as some social impact including religious or ideological habits. According to that there is a strong need to develop appropriate method for distortion identification. The paper fulfil the knowledge gap In the field and seems to be interesting from both scientific and practical point of view. The methodology of the research has been planned properly. Results are shown as clear and corresponding with the conclusions.
